# Race, Zoonoses and Animal Assisted Interventions in Pediatric Cancer

**DOI:** 10.3390/ijerph19137772

**Published:** 2022-06-24

**Authors:** Crina Cotoc, Stephen Notaro

**Affiliations:** 1Medical School and School of Public Health, University of Minnesota, Minneapolis, MN 55455, USA; 2College of Doctoral Studies, University of Phoenix, Phoenix, AZ 85040, USA; snotaro@email.phoenix.edu

**Keywords:** zoonosis, zoonoses, pediatric cancer, pediatric oncology, racial and ethnic minority groups, animal assisted interventions (AAIs), patient, child/children

## Abstract

Emerging evidence accumulates regarding the benefits of animal-assisted interventions (AAIs) in facilitating pediatric cancer treatment and alleviating symptomatology through positive changes in the patients’ emotional, mental, and even physical status. A major concern expressed by healthcare providers and parents in implementing AAIs in hospital settings is the transmission of disease from animals to patients. Immunocompromised children, such as pediatric cancer patients are at increased risk for pet-associated diseases. Furthermore, existing disparities among the racial and ethnic minority groups of pediatric cancer patients can potentially exacerbate their risk for zoonoses. This literature review highlights the most common human infections from therapy animals, connections to the race and ethnic background of pediatric oncology patients, as well as means of prevention. The discussion is limited to dogs, which are typically the most commonly used species in hospital-based animal-assisted therapy. The aim is to highlight specific preventive measures, precautions and recommendations that must be considered in hospitals’ protocols and best practices, particularly given the plethora of benefits provided by AAI for pediatric cancer patients, staff and families.

## 1. Introduction

Pediatric cancer will affect 10,470 children under the age of 15 in 2022 alone in the US, and it is one of the leading causes of death by disease for this population group, alongside congenital malformations and intentional self-harm [1,2]. Despite improved treatments and longer life expectancy, the challenges for pediatric cancer patients are ongoing and lifelong. They struggle daily with debilitating early and late side effects including pharmacogenomic variations with drug exposure, pain, fatigue, psychological problems, fear of cancer recurrences, cognitive difficulties, social problems, and other quality of life aspects [2].

Racial and ethnic groups disparities are present in the rates of the most common types of childhood cancer, with Hispanic children having nearly twice the rate of leukemia as Black children [3].

Like all cancers, gene mutations and alterations lead to uncontrolled cell growth and eventually malignancy. These genetic changes are passed down from parent to child, whose family history, risk factors and prior exposures further enhance racial and ethnic disparities for pediatric cancer. These disparities increase children’s vulnerability to the development of psychological and physical disorders, which may directly or indirectly affect their general clinical condition. The diagnosis and treatment of childhood cancer causes physical and emotional suffering not only for children, but also for parents, caregivers, and their communities [3,4].

There is a growing movement and awareness of treating cancer patients in a holistic approach. Pediatric cancer patients can partake in integrated medicine care, which incorporates conventional medical treatments along with different types of complementary therapies. Animal Assisted Interventions (AAI) is blanket terminology for this type of therapy, including animal-assisted therapy (AAT), animal-assisted activities (AAA), animal-assisted education (AAE) and AAI Resident animals (RA) [5]. Thus, AAI could mitigate the arduous physical and psychosocial symptoms for cancer patients.

Research on human–animal interaction, especially as it relates to pediatric cancer, is limited. Recognition of AAI as an innovative adjuvant therapeutic modality, its safety and efficacy in different environments, including oncologic hospitals, is growing.

One of the main obstacles in a wider implementation of AAIs in the clinical settings is the concern about zoonoses and possibility of infection transmission from the therapy animal to the child patient [4,6]. The evidence of transmitted zoonotic pathogens in this context is scarce. Furthermore, research on susceptibility to zoonoses associated with dogs for minority groups in the US is almost non-existent.

The aim of this review was to synthetize the available data on the most common human infections from therapy animals and their risk factors for pediatric oncology patients, particularly for minority groups of patients. The emerging findings can help improve existing hospital and clinic animal-assisted therapy protocols, with the dogs undergoing health checks for often overlooked pathogens, resulting in safer therapeutic environments for children with cancer, guardians, and staff.

## 2. Materials and Methods

For this paper, we conducted a scientific literature search performed thorough May 2021, examining the following databases: PubMed, PsycINFO, Web of Science, Cumulative Index to Nursing and Allied Health Literature (CINAHL), SCOPUS, Cochrane Library. The search used MeSH terms and included all possible combinations of keywords (with wildcard characters) from the following 4 groups: (1) “animal assisted therapy”, “pet therapy”, “animal assisted intervention”, or “human animal interaction”; (2) “cancer”, “neoplasm”, or “tumor”; (3) “children”, “adolescent”, or “pediatrics”; and (4) “race” or “ethnic background”. We selected articles describing animal-assisted intervention with pediatric cancer patients, as well as articles mentioning racial and ethnic characteristics of pediatric cancer patients. The articles obtained were screened for relevance and selected based on the abstracts and their relevance for the scientific literature. Furthermore, we performed a cited (i.e., forward reference search) and a reference list (i.e., backward reference search) resulting in additional articles. All relevant articles included were in English.

## 3. Results

### 3.1. AAI and Pediatric Cancer

A diagnosis of childhood cancer has debilitating consequences for children, families, and their communities. The demand for a holistic patient-center treatment includes incorporating complementary and alternative adjuvant therapies, such as animal-assisted interventions (AAIs). This terminology encompasses animal-assisted therapy (AAT), animal-assisted education (AAE), animal-assisted activity (AAA) and more [5]. The most common animals used for AAIs in clinical settings are dogs.

Despite the history of human–animal interactions, research on AAI is still in its infancy. Studies on AAI in the pediatric oncologic context are scarce, with only eight known studies conducted internationally. Four of these studies were done in the US (Table 1).

Emerging evidence reports beneficial effects for patients with phycological and physical diseases. The unique human–animal bond is believed to facilitate positive changes in patients’ emotional, behavioral, mental, and physical status [7,8,9,10,11,12,13,14,15,16].

The presence of animals can lead children to have a more favorable perception of the hospital environment, increasing their participation and enhancing their displayed affect [8].

Short-term improvements were found in children’s functional autonomy, such as feeding and nutrition following AAI sessions [9,10]. Results also showed higher heart rates in children participating in AAI compared to children in nonpet therapy groups, but also noticeable pain alleviation, decreased fear and discomfort felt during medical procedures and cancer treatments [8,9,10].

McCullough et al. (2018) noted improvements following AAIs in the initial stages of pediatric cancer treatment, with children experiencing significant reduction in anxiety and parents reporting significant decrease in parenting stress [11].

Hospital staff, parents and guardians recognized positive changes following AAI, including enhanced calmness, motivation and better interpersonal relationships between healthcare professionals and patients, as reported by questionaries of the medical staff in an Italian study and qualitative analysis in a Brazilian study [8,12].

Reported long-term outcomes in children that took part in AAI include improved self-esteem, treatment compliance and motivation. Patients reported being more optimistic about the future, evidenced by a positive correlation between happier mood and higher cortisol levels. Moreira et al. noted a reduction in stress, anxiety, loneliness, and isolation following AAI sessions [12]. On the other hand, McCullough found no significant difference over time between patients exposed to AAI and those who were not [11].

Novel intervention using virtual AAIs sessions (i.e., virtual visits with a dog or a cat through letter writing and pictures) promoted well-being, improved connection, and friendship among pediatric cancer patients [13].
ijerph-19-07772-t001_Table 1Table 1AAI and Pediatric Oncology Studies.Author/YearOutcomesParticipant DemographicsInstrumentsMain FindingsBouchard et al., 2004 [9]Satisfaction with the program and of quality of care Functional independenceNourishmentPhysical exercise Age: 3 to 13 years old 46% male, 54% femaleSelf-administered questionnairesPotentially beneficial role of animal therapy on the physical dimensions, social, emotional and coping, and self-esteemCaprilli and Messeri, 2006 [8]Children’s ParticipationChildren’s PleasureParents’ Level of SatisfactionStaff Members’ Level of SatisfactionHospital-wide infection rate before and after AAIAge: 4–12 years oldSelf-assessment questionnaires and behavioral scalesSome beneficial effects of AAI on children: a better perception of the environment and a good interaction with dogs. A total of 94% of parents had positive perception of AAI. The medical staff needed more information about safety. The presence of infections in the wards did not increase after AAIs.Chubak and Hawkes, 2016 [14]Self-reported moodDisplayed affectAmount of touch, BP/HR, Salivary cortisolAge: 5 years of age or older 56% males, 44% femalesSurveysSubstantial variation in practice of AAI with pediatric oncology patientsGillespie and Neu, 2020 [13]Experiences of YAPS participants over time and AAT as alternative intervention to the traditional form of AATAge: 7–16 years old7 males, 8 females14 Caucasians, 1 HispanicInterviewsA virtual letter writing program can provide pediatric oncology patients a source for connection, friendship, shared experience, fun, and a way to process the cancer experience with a dog or cat pen pal who has also faced cancer or serious medical treatment.Kaminski et al., 2002[15]Self-reported moodDisplayed affectAmount of touch, Heart rate/ blood pressure Salivary cortisolAge: 5 years of age or older 56% males, 44% femalesInterviews and videotape assessmentsMeasures of cortisol levels, BPSignificantly increase in positive affects in AAI groupsIncrease in heart rate in AAI groupsNo significant differences between mood between groups McCullough et al., 2018[11]Stress and anxiety in children and parentsQuality of lifeParental stressBP and HRAge: 3–17 years oldWhite: 67.9%, African American: 7.5%,Hispanic/Latino: 14.2%, Other: 5.7%Self-report, STAI, PedsQl, PIP questionnaires, physical measurementsAnimal-assisted interventions may provide certain benefits for parents and families during the initial stages of pediatric cancer treatmentMoreira et al., 2016[12]Perception of professionals and legal guardians of children and adolescents with cancer regarding Assisted Therapy with DogsAge: 4–7 years old Semi-structured interviewsRecognized benefits of AAIParents do not understand the objectives and therapeutic applications of AAIPotential for AAI to promote better health for pediatric cancer patientsSilva and Osorio, 2018[16]Efficacy and safety of a protocol for animal assisted therapy; Stress, pain, mood, anxiety, depression, Q of Life, HR, BPAge: 6–12 years old41.7% males,58.3% femalesCSSI, QLES, CDI, Adapted BMS, FPS, AAT Assessment Questionnaire, STAI, Sociodemographic and Clinical Identification QuestionnairesDecrease in pain, irritation, stress and tendency to improvement on depressive symptoms following AAIImprovement in anxiety, mental confusion and tension for caregiversEffective program for pediatric oncologic patients mainly in outpatient unitsBP = Blood pressure; STAI = State-Trait-Anxiety-Inventory; PedsQl = Pediatric Quality of Life Inventory; PIP questionnaires = Pediatric Inventory for Parents questionnaires; CSSI = Child Stress System Inventory; QLES = Quality of Life Evaluation Scale; CDI = Child Depression Inventory; Adapted BMS = Adapted Brunel Mood Scale; FPS = Faces Pain Scale; AAT = Animal-Assisted Therapy Assessment Questionnaire.

AAIs can contribute to improving the overall quality of life in children experiencing cancer, by decreasing pain and fear, providing a much-needed distraction from the gruesome reality of cancer treatments and procedures, lowering emotional distress, and facilitating symptom management [4,6,12].

### 3.2. Zoonoses Associated with Dogs

Despite approval and recognition of the beneficial effects of AAI, parents/guardians requested additional safety issue facts and information [8]. Pediatric oncologic patients due to their immunodeficiency status are at higher risk of developing disease or complications from zoonotic diseases.

Furthermore, parents and staff lacked adequate knowledge on the implementation of AAIs, as well as insufficient understanding of the therapeutic objectives of animal assisted therapy with pediatric cancer patients [12].

In this context, the close interactions between humans and animals increases the risk factors and can facilitate the transmission of potentially pathogenic microorganisms from animals to patients. Exposure can occur through inhalation and contact with skin, eyes and mucous membranes. Transmission occurs through direct contact, inhalation of aerosols, infected saliva, contaminated urine or feces, or contact with contaminated objects [17,18,19].

Common transmissible diseases associated with dogs include viral infections (Norovirus infections, Rabies), bacterial infections (*Bordetella bronchiseptica*-associated disease, Brucellosis, Campylobacterosis, Capnocytophagosis, Coxiellosis, Cryptosporidiosis, Infections with pathogenic *E. coli*, Leptospirosis, Methicillin resistance *Staphylococcus aureus*/MRSA, Pasteurellosis, Salmonellosis, Staphylococcal pyoderma, Tularemia cutaneous, Yersiniosis enterocolitica), fungal infections(Ringworm), parasites (Echinococcosis, Giardiasis, Mange) and Visceral larva migrans [17,18,19,20] (Table 2). 

For the majority of zoonoses, effective treatments are available. Basic preventive care (e.g., internal and external parasite control, vaccination) protects both canine and human health and is further enhanced by animal and environmental management control. Epidemiologic studies on the topic suggest that the occurrence of dog associated zoonotic disease is low overall [19]. Results from an Italian study showed that 31.6% of the animals involved in AAI over a two-year period positive for pathogens, with 21.7% of the dogs harboring potentially dangerous zoonotic parasites (*Ancylostomatidae*, *Eucoleus aerophilus*, *Toxocara canis*, *Giardia duodenalis*, *Nannizzia gypsea* and *Paraphyton mirabile*) [7]. Availability of educational resources and open communication about zoonotic diseases for medical staff, dog handlers and patients is critical and ensures good practices. It also builds trust between patients, AAI teams and medical staff.

The surveillance of zoonotic pathogens in the context of AAI is necessary for ensuring a safe environment in the clinical setting, for patients, staff and families. Prevention methods for zoonotic diseases vary for different pathogens; however, standard preventive measures (vaccination, serology testing) are recognized as effective tools in reducing the risks of exposure and transmission.

### 3.3. Susceptibility to Zoonoses by Race

Cancer is the leading cause of death by disease past infancy among children in the United States. Among children ages 0 to 14 years, it is estimated that in 2021, 10,500 will be diagnosed with cancer and 1190 will die of the disease [3].

Within the pediatric population, regarding racial and ethnic minority children, we found that compared to non-Hispanic White childhood cancer patients, Black and Hispanic patients had worse survival for all cancers combined, leukemias and lymphomas, brain tumors, and solid tumors [21,22,23,24,25].

Black and Hispanic patients had a higher risk of death compared to White, non-Hispanic patients. Black patients had a higher risk of death at 5 years after diagnosis compared to non-Hispanic white patients, with a 5-year relative survival rate of 69.8% for Black patients, 72.9% for Hispanic patients and 77.6% for White patients [21,22,23]. Similar differences were found when comparing 12.5-year survival rates [24]. Marcotte found the incidence of leukemias to be higher in White children when compared to both Black and Asian/Pacific Islander children [25].

Despite similar reported survival rates for Hispanics, African Americans, Asians and White children, gaps in understanding disparities in cancer still exist [26,27]. Racial and ethnic disparities in childhood CNS tumor survival appear to have their roots at least partially in post-diagnosis factors, potentially due to the lack of access to high quality care, leading to poorer overall outcomes [28,29]. These disparities include differences or delays in treatment. Black race, Hispanic ethnicity, lack of private insurance, and adolescent/young adult age are most often associated with these poorer outcomes [26]. Additional disparities include impaired access to care and clinical trials, differences in cancer biology, treatment non-adherence, language barriers, and implicit racial bias. Although socioeconomic factors may account for a large proportion of disparities seen, the causes of disparities are complex and interconnected and still need to be better understood [27,28]. Further studies on how systemic racism and oppression impact pediatric cancer are needed.

Prior research suggests that ethnicity is a crucial factor shaping disease knowledge [30]. Hispanics/Latinos have 30% less knowledge on rabies compared to non-Latino Whites counterparts. Language and cultural barriers, the lack of available educational materials in languages other than English for transmissible infectious diseases, all play a role. Furthermore, African American people have lower knowledge on rabies than their White counterparts. In this case, trust in the information source was a key factor, with African Americans having low or diminished trust in public health authorities. Women also scored higher on knowledge about rabies than males [31].

Health disparities are a heavy burden on the US healthcare system, with racial and ethnic differences in chronic disease morbidity and mortality well documented. Non-Hispanic Blacks and Hispanics exhibit odds 1.7 times and 2.8 times higher than those of non-Hispanic Whites for contracting enteric pathogens [32]. Non-Hispanic Blacks were almost twice as likely as non-Hispanic Whites to be seropositive for Helicobacter pylori, and Mexican Americans were 2.2 times more likely to be seropositive than non-Hispanic Whites. This higher prevalence of infection for minority groups is also reported for Toxoplasma gondii, Hepatitis B virus, Hepatitis C virus, Herpex Simplex2 [32].

These disparities are inherited by minority children, and hence by minority pediatric cancer patients. Combined with the immunocompromised status as a result of cancer treatments, the prevalence for infections for minority pediatric cancer patients increases considerably.

### 3.4. Existing Hospital Protocols for AAIs

Perhaps in part due to broad media claims of patients benefits, canine-assisted interventions are becoming more popular in hospital settings [33]. States including Illinois (Ill. Admin. Code tit. 77 § 250.890) have established policies for AAI based on CDC recommendation regarding establishing a department in charge, sanitation and infection control, certification of training, patient screening for participation, areas permitted, length of sessions, type of animals, policies for incidents, and patient consent [34]. Despite the existence of these proposed guidelines for AAI in hospitals, there are significant differences in infection control policies across these groups [35]. Additional research is needed to investigate whether therapy animals can serve as pathogen vectors, from being contaminated by contact with one patient, and then transmitting these pathogens to another patient, leading to pathogen exchange. This is critical to test because many patients served by these therapy animals have a compromised health status and may be at higher risk of infection compared to the general population [36].

Chuback and Hawkes (2016) reported on the variability of practices and protocols used across institutions. These differences in protocols and intervention designs could potentially compromise the outcomes, safety and generalizability of findings [14].

Current guidelines recommend that the following 10 health-related factors be evaluated at each life stage for a canine patient: lifestyle effect on the patient’s safety, zoonotic and human safety risk, behavior, nutrition, parasite control, vaccination, dental health, reproduction, breed-specific conditions, and a baseline diagnostic profile [37,38].

Findings of AAI studies with pediatric cancer patients show a lack of standardization of the number of canines, duration, and frequency of sessions, the executed activities, and the safety measures for the animals and cancer patients. Lefebvre et al. (2008) advocated the need for universal, consensual and collaborative guidelines that represents the interests of the stakeholders in the pediatric cancer arena, which provides specific recommendations to minimize both injuries and the transmission of infectious organisms to and from therapy animals [37,38,39]. The guidelines developed recommended a less-than-rigorous screening protocol to identify animal carriage of specific pathogens (including group A streptococci, Clostridium difficile, vancomycin-resistant enterococci, and MRSA). Exceptions were made for cases when the animal interacted directly with a human carrier or if the dog is linked to an outbreak of infectious diseases [39].

Silva and Osorio (2018) proposed and implemented an effective AAI protocol for outpatient pediatric oncology patients in Brazil. Patients were screened for severe infections, such as infection by resistant bacteria (Staphylococcus aureus or coagulase-negative oxacillin-resistant S. aureus, vancomycin-resistant Enterococcus, or cefepime-resistant and/or meropenem-resistant Gram-negative bacillus), or suspected or confirmed infection with Clostridium difficile; no details are provided on what infections the therapy animals were screened for [16].

Santaniello et al. (2020) reported on the presence of P. multocida in dogs performing AAI, highlighting the potential risk of this infection being transmitted from therapy animals to vulnerable individuals, including immunocompromised patients [40]. Furthermore, results of a systematic review emphasize the need for mandatory microbial control of the therapy animals, along with strong hygienic rules, considering the evidence of increased risk of zoonoses associated with dogs for ESKAPE pathogens (*Enterococcus faecium*, *Staphylococcus aureus*, *Klebsiella pneumoniae*, *Acinetobacter baumannii*, *Pseudomonas aeruginosa*, *Enterobacter* spp.) [41].

Studies show a lack of effective educational campaigns and open communication networks between hospital infection control departments and therapy animal handlers regarding infection risk [36]. Expanding and improving infection control measures for AAIs in cancer hospitals and clinics is critical to obtain a positive balance of the benefits and risks for all stakeholders [42,43].

## 4. Discussion

Historically, African Americans have had the highest death rate and lowest survival rate of any racial or ethnic group for most cancers. Although there have been improvements in recent years, persistence of systemic racism, oppression, lower socioeconomic status, impaired access to health insurance and adequate medical care services support existing health disparities, manifested in African Americans still being the populational group with the highest cancer rates. Non-Hispanics have also a higher risk of death compared to their White counterparts [1,3].

Pediatric cancer is one of the few cancer types where the discrepancies in cancer incidence are lower. Black children experience decreased incidence of acute lymphoid leukemia compared to Whites, and this decreased incidence was strongest at ages 1 through 7 years. Hispanic children have a decreased overall incidence of Hodgkin lymphoma and astrocytoma but experience increased risk of acute lymphoblastic leukemia compared to non-Hispanic Whites. Substantially decreased risk across many tumor types was observed for Asian/Pacific Islanders and American Indian/Alaska Natives pediatric cancer patients [3,21,22,23,24,25].

Despite these encouraging statistics, health disparities impacting minority groups are also affecting children experiencing cancer. A cancer diagnosis has major repercussions not only for the child who is diagnosed, but also for his immediate family and caretakers. Systemic oppression, racism, financial burdens all play a role in increasing the disease burden for disadvantaged and minority groups of patients.

Furthermore, prior evidence suggests a higher susceptibility to infections for Hispanics and Blacks. A Canadian study showed that therapy dogs visiting hospitals have almost five times higher odds of carrying MRSA than therapy dogs who visit other locations, such as schools [43]. Compounding the immunodeficient health status of pediatric cancer patients, their risk for zoonotic diseases increases. Mitigating the zoonoses risks requires all AAI stakeholders in a clinical setting to collaborate and have accurate and timely information on infection diseases [19].

Previous studies have not reported any serious negative impacts of AAI, and most report at least some positive effect on the patients and families. Several studies speak of the benefits of AAI.

Chia-Chun et al. [44], in a study of children’s systolic blood pressure (SBP) when measuring SBP before and after AAI, reported decreased levels from before to during and continued after the AAI interaction [45]. Lindstron-Nilsson et al. (2020) found the well-being of children increased to very good after AAI and reported the hospital stay as better with 93% of the children assessing their interaction with the dog as very good [45]. Zeblisky and Jennings (2016) reported similar results at a large children’s hospital. Data from a 9-year study indicated a 93% percent positive change in the mood of patients [46]. Avila-Alverez and Pardo-Vasquez (2020) concur as children and parents gave AAI their highest satisfaction rating with a significant improvement in the child’s mood [47]. In a survey of parents and staff at a children’s hospital, Uglow (2019) reported that out of 200 respondents, 100% recommended expanding the services across the U.K. and had no concerns of the dogs being present, their behavior or cleanliness [48].

Concerns of disease transmission may be better addressed with the active involvement of veterinarians. Currently, in many hospital settings, the veterinarian role is limited to requests made by the pet owner for participation based on hospital policies. Expanding that role so the veterinarian is an active member of a team that includes hospital staff, the individual responsible for care of the animal would enhance the ability to focus on the well-being of the pet, possibly mitigation of the spread of disease and enhance the ability to maximize the benefit of the AAI [37,38]. To assist veterinarians working with animals participating in AAI, products such as wellness health care checklists could facilitate communications between the veterinarian and client relating to pets used in AAI.

As AAI are becoming more and more popular in hospital settings, the active involvement of veterinarians could play a vital role in working alongside hospital administrative and clinical staff, and the AAI handlers in developing safeguards to minimize risk and maximize benefits to venerable humans and protect therapy dog welfare.

## 5. Conclusions

A successful AAI hospital protocol and disease prevention program relies on cancer patients, parents, therapy animals’ handlers and staff receiving accurate, timely advice and information on risk reduction for zoonoses. Given the health benefits of AAI and the vulnerability of pediatric cancer patients, well outlined infection control measures must be included in any hospital/clinic protocol for therapy animals. Minority children, particularly the immunocompromised cancer patients, are at increased risk for zoonotic infections, so additional safeguards should be considered.

These safeguards include specific consideration for the most common zoonoses associated with animal-assisted therapy dogs. Screenings and preventive vaccinations should be discussed for the zoonoses associated with dogs, with moderate risk of transmission. Considering their wealth of knowledge and expertise on zoonoses, veterinarians should be consulted and included in the teams that develop clinical protocols for animal assisted interventions in any oncologic treatment and care setting.

## Figures and Tables

**Table 2 ijerph-19-07772-t002:** Common Zoonoses Associated with Dogs.

Disease	Pathogen	Risk of Transmission	Symptoms in Humans **	Treament **
**Viral**				
Norovirus infections	*HuNoV*	Low	Viral gastroenteritis	No specific treatment
Rabies	*Rabies virus/Lyssavirus genus* within the *Rhabdovirus* family	Low	Flu-like symptoms initially; Progression to cerebral disfunction, agitation, confusion and death.	PEP */Potent rabies vaccine
**Bacterial**				
*Bordetella bronchiseptica*- associated disease	*Bordetella bronchiseptica*	Moderate	Dry cough, sinusitis, bronchitis, pneumonia	Antibiotic treatment, antitussives, bronchodilators
Brucellosis	*Brucella Canis*	Low	Flu-like symptoms, septicemia, cardiac and neurological symptoms, infertility	Prolonged antibiotic treatment
Campylobacterosis	*Campylobacter jejuni* and *Campylobacter upsaliensis*	Low	Gastroenteritis/ stomach flu	Specific antibiotic treatment in severe cases
Capnocytophagosis	*Capnocytophaga canimorsus*	Low	Flu-like illness, skin rash, septicemia	Antimicrobial therapy
Q fever/Coxiellosis	*Coxiella burnetii*	Low	Mild flu-like symptoms	None or Tetracyclines in severe cases
Cryptosporidiosis	*Cryptosporidium oocysts*/protozoan parasite	Low	Gastroenteritis, watery diarrhea, vomiting	Nitazoxanide
Infections with pathogenic *E. coli*	*Escherichia coli*	Moderate	Cholecystitis, bacteremia, cholangitis, urinary tract infection (UTI), and traveler’s diarrhea, and other clinical infections such as neonatal meningitis and pneumonia.	Rest, fluids. Combination therapy with antibiotics plus antianaerobe in severe infections
Leptospirosis	*Leptospira spirochete*	Low	High fever, headache, chills, muscle aches, vomiting, jaundice, red eyes, abdominal pain, diarrhea, rash	Antibiotics (doxycycline or penicillin)
MRSA	Methicillin-resistant *Staphylococcus aureus*	Moderate	Staph skin infections and fever.	Certain antibiotics, surgery to drain abscesses
Pasteurellosis	*Pasteurella*	Moderate	Local wound infection	Broad spectrum antibiotics
Salmonellosis	*Salmonella*	Low	Diarrhea, stomach cramps, fever, nausea, vomiting, chills, bloody stool	Dehydration treatment, anti-diarrheals and antibiotics
Staphylococcal pyoderma	*Staphylococcus intermedius*	Moderate	Skin infection	Combination antibiotic therapy
Tularemia cutaneous	*Francisella tularensis*	Low	Skin ulcer, swollen and painful lymph glands, fever, chills, headache, exhaustion, eye swelling, gastrointestinal, flu-like symptoms, muscle pain, pneumonia	Intravenous antibiotic therapy, oral antibiotics
*Yersiniosis enterocolitica*- associated disease	*Yersinia enterocolitica*	Low	Fever, abdominal pain, bloody diarrhea	Antibiotics in severe cases
**Fungal**				
Ringworm	*Microsporum* spp., *Trichophyton* spp.	Moderate	Itchy skin, ring-shaped rash, red and scaly skin, hair loss	Clotrimazole, Miconazole, Lamisil, Ketoconazole, Fluconazole, etc.
**Parasites**				
Echinococcosis	*Echinococcus granulosus* and *Echinococcus multilocularis*	Low	Abdominal pain, liver cyst, bloody sputum, chest pain and cough (lung cyst), anaphylaxis	Antiparasitic therapy combined with either surgical resection of the cyst or percutaneous aspiration and instillation of scolicidal agents
Giardiasis	*G. duodenalis*	Moderate	Watery stools, fatigue, stomach cramps, bloating, nausea, weight loss	Metronidazole, Tinidazole, Nitazoxanide
Mange	Mites	Moderate	Severe itching, skin blisters and bumps	Permethrin cream, Ivermectin, Crotamiton
Visceral larva migrans	*Toxocara Canis*	Low	Cough, fever, hepato-spleno-lymph adenopathy, pulmonary infiltrates, CNS involvement	Albendazole, Mebendazole, Diethylcarbamazine, Ivermectin

* PEP = post-exposure prophylaxis; HuNoV = Human Norovirus. ** Data from Refs. [17,18,19].

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
