# Peer review of "Race, Zoonoses and Animal Assisted Interventions in Pediatric Cancer"

_ijerph, 2022, doi:10.3390/ijerph19137772_

Round 1

Reviewer 1 Report

This literature review highlights the most common human infections from  animals involved in animal assisted interventions (AAIs). This review is well written and add specific preventive measures, precautions and recommendations that must be considered in hospitals’ protocols and  best practices, particularly given the plethora of benefits provided by AAIs for pediatric cancer patients, staff and families.

In my opinion, this review may be accepted in the present form.

Author Response

We would like to thank Reviewer 1 for their time and attention in reviewing our manuscript and for their useful comments and recommendations. 

Reviewer 2 Report

1, Line 27, is the leading cause of death by disease this population group leading cause of death disease for this population group

Would the authors please revise the sentence to more comprehensively include all-cause of leading mortality in these age groups, including trauma, homicide, and suicide, that has been updated by the CDC?

2. Lines 28-31, the authors need to specifically describe the factors related to the pediatric populations, including but not limited to the longer life expectancy

3. The first paragraph in the introduction section should include a brief description of Animal-assisted interventions.

4. Lines 104-105, the authors are encouraged to explain the evidence of improved patient and staff interactions, and whether this improvement was quantified according to a specific questionnaire.

5. The authors are encouraged to explain the reason that despite the considerable rate of the zoonotic disease, the animals were not screened and or treated before entering the trials.

6. Line 168, would the authors clarify, whether 2021 is a typo? otherwise, the year, and the according data need to be corrected.

7. Line 269, have had have should be edited

Author Response

We would like to thank Reviewer 2 for their time and attention in reviewing our manuscript and for their useful comments and recommendations.  We believe that incorporating their suggestions has greatly improved the manuscript and helped to refine its content. 

Reviewer: 2 
Comments to the Authors 

Page 1, Line 27: “is the leading cause of death by disease this population group leading cause of death disease for this population group. Would the authors please revise the sentence to more comprehensively include all-cause of leading mortality in these age groups, including trauma, homicide, and suicide, that has been updated by the CDC?”

*Thank you for this feedback.  We have revised the phrase and amended the leading causes of death by disease to include self-harm (suicide) and congenital malformations. We have not included trauma and homicide as they are not causes of death by disease. We have also updated our references to include the updated CDC statistics. Page 1, Lines 27-28 

Page 1, Lines 28-31: “the authors need to specifically describe the factors related to the pediatric populations, including but not limited to the longer life expectancy”

*We appreciate this observation. Aligning with the suggestion to include factors related to the pediatric populations, we have changed this sentence to read “Despite improved treatments and longer life expectancy, the challenges for pediatric cancer patients are ongoing and lifelong”.  Additionally, we have added “pharmacogenomic variations with drug exposure” to the list of side effects.  Page 1, Lines 29-31

Page 2, Lines 48-51: The first paragraph in the introduction section should include a brief description of Animal-assisted interventions.

* We thank you for this comment. We have included a definition for AAI according to American Veterinary Medical Association. We have also updated our references.  Page 2, Lines 48-50

Page 3, Lines 104-105: the authors are encouraged to explain the evidence of improved patient and staff interactions, and whether this improvement was quantified according to a specific questionnaire.

*Thank you. We have changed this line to be more clear: “as reported by questionaries of the medical staff in an Italian study and qualitative analysis in a Brazilian study.” Page 3, Lines 108-109

Comment 5: The authors are encouraged to explain the reason that despite the considerable rate of the zoonotic disease, the animals were not screened and or treated before entering the trials.

*Thank you.  We agree with this observation, however we addressed this in the paper mentioning less than rigorous guidelines for screening protocols. No additional reasons are given in the literature why the dogs are not screened for additional zoonotic diseases.   Page 10, Lines 248-252

Page 8, Line 168: would the authors clarify, whether 2021 is a typo? otherwise, the year, and the according data need to be corrected.

*Thank you.  We have doubled checked our source and according to the National Cancer Institute, the estimated pediatric diagnoses in 2021 are 10,500, including 1,190 deaths. No data for 2022 is available yet. Page 8, Lines 173-174

Page 9, Line 269: have had have should be edited

*Thank you very much for catching this typo. We have corrected it. Page 10, Line 274

Reviewer 3 Report

This is a very interesting review article that focuses on animal-assisted interventions (AAIs) used in patients with pediatric cancer, especially children from racial and ethnic minority groups. Several factors were taken into consideration, including AAI in clinical settings, related zoonoses, racial differences, and hospital protocols. This topic is also novel, as few papers investigated relevant studies. The content is comprehensive. Some points (such as the involvement of veterinarians and pathogen exchange from one patient to another through animals) provide helpful information for clinical practice.

Minor comments:

Some typos: Page 10 line 265, 267 

Please provide the reference numbers in Table 1.

Author Response

We would like to thank Reviewer 3 for their time and attention in reviewing our manuscript and for their useful comments and recommendations.  We believe that incorporating their suggestions has greatly improved the manuscript and helped to refine its content. 

Reviewer: 3 
Comments to the Authors 

Overall feedback: 
This is a very interesting review article that focuses on animal-assisted interventions (AAIs) used in patients with pediatric cancer, especially children from racial and ethnic minority groups. Several factors were taken into consideration, including AAI in clinical settings, related zoonoses, racial differences, and hospital protocols. This topic is also novel, as few papers investigated relevant studies. The content is comprehensive. Some points (such as the involvement of veterinarians and pathogen exchange from one patient to another through animals) provide helpful information for clinical practice.

Minor comments:

Some typos: Page 10 line 265, 267

Please provide the reference numbers in Table 1.

*Thank you very much for the kind comments and useful feedback.

We have corrected the typos on Page 10, Lines 265, 267 and added reference numbers in Table 1.